# Genes *ScBx1* and *ScIgl*—Competitors or Cooperators?

**DOI:** 10.3390/genes11020223

**Published:** 2020-02-20

**Authors:** Anna Wlazło, Magdalena Święcicka, Marek D. Koter, Tomasz Krępski, Leszek Bolibok, Anna Stochmal, Mariusz Kowalczyk, Monika Rakoczy-Trojanowska

**Affiliations:** 1Department of Plant Genetics, Breeding and Biotechnology, Institute of Biology, Warsaw University of Life Sciences, 02-787 Warsaw, Poland; annawlazlo@outlook.com (A.W.); magdalena_swiecicka@sggw.pl (M.Ś.); marek_koter@sggw.pl (M.D.K.); tomasz_krepski@sggw.pl (T.K.); 2Department of Silviculture, Institute of Forest Sciences, Warsaw University of Life Sciences, 02-787 Warsaw, Poland; leszek_bolibok@sggw.pl; 3Department of Biochemistry and Crop Quality, Institute of Soil Science and Plant Cultivation, State Research Institute, 24-100 Puławy, Poland; asf@iung.pulawy.pl (A.S.); mkowalczyk@iung.pulawy.pl (M.K.)

**Keywords:** rye, gene expression, gene silencing, benzoxazinoid biosynthesis

## Abstract

Two genes, *Bx1* and *Igl*, both encoding indole-3-glycerol phosphate lyase (IGL), are believed to control the conversion of indole-3-glycerol phosphate (IGP) to indole. The first of these has generally been supposed to be regulated developmentally, being expressed at early stages of plant development with the indole being used in the benzoxazinoid (BX) biosynthesis pathway. In contrast, it has been proposed that the second one is regulated by stresses and that the associated free indole is secreted as a volatile. However, our previous results contradicted this. In the present study, we show that the *ScIgl* gene takes over the role of *ScBx1* at later developmental stages, between the 42nd and 70th days after germination. In the majority of plants with silenced *ScBx1* expression, *ScIgl* was either expressed at a significantly higher level than *ScBx1* or it was the only gene with detectable expression. Therefore, we postulate that the synthesis of indole used in BX biosynthesis in rye is controlled by both *ScBx1* and *ScIgl*, which are both regulated developmentally and by stresses. In silico and in vivo analyses of the promoter sequences further confirmed our hypothesis that the roles and modes of regulation of the *ScBx1* and *ScIgl* genes are similar.

## 1. Introduction

Benzoxazinoids (BXs) are secondary metabolites synthesized by many species from the Poaceae family and, sporadically, by several dicots. The first step, a branch point in BX biosynthesis, that occurs in chloroplasts is the conversion of indole-3-glycerol phosphate (IGP) to indole. The next reactions take place in other cell compartments: four subsequent reactions leading to the synthesis of 2,4-dihydroxy-1,4-benzoxazin-3(4H)-one (DIBOA) (in endoplasmic reticulum) followed by the glycosylation of DIBOA to 2-O-β-glucoside (GDIBOA) (stored in vacuoles) and O-methylation leading to 2,4,7-trihydroxy-1,4-benzoxazin-3-one glucoside (GTRIBOA), 2,4-dihydroxy-7-methoxy- 1,4-benzoxazin-3(4H)-one (DIMBOA) glucoside (GDIMBOA), and 4,7-dimethoxy-2-{[3,4,5-trihy- droxy-6-(hydroxymethyl)oxan-2-yl]oxy}-3,4-dihydro-2H-1,4-benzoxazn-3-one glucoside (GHDMBOA) (in the cytoplasm). These glucosides are stored in the vacuoles. After hydroxylation, GDIBOA and GDIMBOA are converted into DIBOA and DIMBOA, respectively, and released from the vacuoles into the cytosol (Figure 1) [1,2,3,4,5,6]. In rye, ten orthologs of maize *Bx* genes, namely, *ScBx1–ScBx7*, *ScIgl*, *Scglu*, and *ScGT* (the last one corresponding to maize *Bx8* and *Bx9*), have been isolated in the last decade [7,8,9,10,11]. In addition to *Bx1*, two other genes, namely, *Igl* and *TSA*, control the transformation of IGP to indole. Indole produced in the reaction catalyzed by the BX1 enzyme encoded by the *Bx1* gene is used in BX biosynthesis. *TSA* encodes the Alpha subunit of tryptophan synthase (TSA), which catalyzes the same reaction; however, its product, indole, serves as the penultimate intermediate in the formation of tryptophan. In barley and several other species belonging to Poaceae, tryptophan is used in the biosynthesis pathway of gramine, a group of compounds playing roles in plant defense reactions similar to those of BXs [12]. Finally, *Igl* encodes indole-3-glycerol phosphate lyase (IGL), which produces free indole as a response to pest feeding or volicitin, an elicitor molecule that can mimic larval feeding [2,13]. Recently, a group of Groszyk et al., which included the last three authors of this article, showed that the expression of the *ScBx1* gene increases under native conditions until the third day after germination and then decreased rapidly, becoming undetectable on the 21st day [14]. This situation was found to change completely after infection with *Barley stripe mosaic virus* (BSMV)-based vector; that is, the expression of *ScBx1* increased nearly 40-fold. The authors concluded that *ScBx1* is not only regulated developmentally, but also activated by biotic stress, namely, virus infection. The authors also found that, despite the lack of expression of the *ScBx1* gene from the third week of cultivation, BXs (HBOA (2-hydroxy-1,4-benzoxazin-3-one), DIBOA, DIMBOA, GDIMBOA, and MBOA (6-methoxy-benzoxazolin-2(3H)-one)) were still synthesized. They assumed that another gene, most probably *ScIgl*, may control the first step of rye BX biosynthesis at later developmental stages.

To resolve this issue of the roles of the *ScBx1* and *ScIgl* genes and in which manner they cooperate or compete, we performed a detailed expressional analysis of these genes in untreated rye plants and, additionally, promoter analysis in plants infected with brown rust. To provide final confirmation of the participation of the *ScIgl* gene in BX biosynthesis control, in the experiments we included plants with silenced *ScBx1* gene that were previously obtained and characterized in terms of DIBOA content [14].

## 2. Materials and Methods

### 2.1. Plant Materials

The plant material consisted of three rye inbred lines: L318 bred at the Department of Plant Genetics, Breeding and Biotechnology, Warsaw University of Life Sciences; and D33 and D39, derived from the Polish breeding company Danko Plant Breeders Ltd. The criterion for selecting inbred lines was the content of BX measured previously after the period of natural vernalization [16]. L318 was characterized as a line with high, D33 with intermediate, and D39 with low BX content (Appendix A). Seeds of these three rye lines were germinated on Petri dishes with wet cotton wool and kept in a plant growth room (16/8 h photoperiod, 22 °C). After 4 days, the seedlings were transferred to pots with peat substrate and grown under the same conditions.

The aerial parts of plants were collected at the following time points: 14, 21, 28, 42, 70, and 77 days after germination (dag). At each time point, the tissues were divided into equal parts: one part was assigned to RNA isolation and the other to biochemical analysis of BX content. The tissues assigned to biochemical analysis were lyophilized (Alpha model 2-4 LDplus; Martin Christ Gefriertrocknungsanlagen GmbH, Osterode am Harz, Germany).

### 2.2. RNA Isolation

RNA was isolated using a GeneMATRIX Universal RNA Purification Kit version 1.2 (Eurx, Gdańsk, Poland). Shortly thereafter, 100 mg of frozen tissue was ground in liquid N_2_, and total RNA was isolated in accordance with the manufacturer’s protocol. The RNA was diluted in 40 μL of RNase-free water. The RNA integrity and concentration were measured using a NanoDrop 2000 spectrophotometer. Then, to avoid genomic DNA contamination Turbo DNase (Thermo Fisher Scientific, Waltham, MA, USA) was used, in accordance with the manufacturer’s instructions.

### 2.3. cDNA Synthesis

The High-Capacity cDNA Reverse Transcription Kit (Thermo Fisher Scientific) was used to synthesize cDNA from isolated RNA. The total volume of the reaction mixture was 20 μL, which contained 2 μg of template RNA, 2 μL of 10× RT buffer, 0.8 μL of 25× dNTP mix, 2 μL of 10× RT random primers, 1 μL of MultiScribe reverse transcriptase, and 4.2 μL of nuclease-free water. The reaction was performed at 25 °C for 10 min, 37 °C for 120 min, and 85 °C for 5 min, after which the obtained cDNA was diluted. cDNA from virus-silenced rye plants (cv. Stach F_1_) was obtained previously by Groszyk et al. [14].

### 2.4. qRT-PCR

The qRT-PCR reaction for the two genes, *ScBx1* (GenBank: KF636828.1) and *ScIgl* (GenBank: MN120476), was performed using LightCycler 96 Real Time System (Roche, Basel, Switzerland), in accordance with the following program: pre-incubation at 95 °C for 600 s; 32 cycles of three-step amplification at 95 °C for 10 s, 58 °C (*ScBx1*) or 55 °C (*ScIgl*) for 10 s, and 72 °C for 15 s; and then melting at 95 °C for 10 s, 55 °C for 60 s, and 97 °C for 1 s. On the basis of a literature search, a few commonly used reference genes were chosen (e.g., actin, glyceraldehyde phosphate dehydrogenase (GADPH), cell division control protein, AAA-superfamily of ATPases—Ta54227). Primers of all selected reference genes were tested and finally the barley actin gene, *HvAct* (GenBank: AY145451), was selected as the most stably expressed gene and used as an internal control reference gene in our experiments. The total volume of the reaction mixture was 20 μL, which contained 4 μL of cDNA, 1 μL of gene-specific primers (5 μM) each, 4 μL of RNase-free water, and 10 μL of FastStart Essential DNA Green Master (Roche). The primer sequences are listed in Table 1. Relative quantification of *ScBx1* and *ScIgl* expression level was calculated using ΔΔCt method. The expression level of studied genes was normalized to the expression level of reference gene (*HvAct*). The qRT-PCR reaction was performed in three biological (four plant per replicate) and two technical replicates.

### 2.5. Analysis of ScBx1 and ScIgl Expression in Plants with Silenced ScBx1

The qRT-PCR was performed on the cDNA of rye plants, cv. Stach F_1_, with virus-induced silenced *ScBx1* gene as previously obtained by Groszyk et al. [14]. For the analysis, plants with completely or considerably silenced *ScBx1* gene at a given time point, namely, 14 or 21 days post-inoculation (dpi), were selected. These were as follows: 5 plants designated as #1, #3, #5, #7, and #9 at 14 dpi and 9 plants designated as #2, #5, #6, #8, #10, #11, #12, #13, and #14 at 21 dpi. The conditions of qRT-PCR including primer sequence were the same as described above. Three technical replicates for each individual plant used in the experiment were performed. The relative expression level of *ScBx1* and *ScIgl* gene was normalized to their expression level in plants inoculated with BSMV:α,β_(-)_,γ_(*PDS*)_ vector assumed as 1.

### 2.6. In Vivo Analysis of Promoter Sequences

For the analysis of proteins related to biotic stress (infection by brown rust, a disease caused by the obligate biotrophic basidiomycete fungus *Puccinia recondita* f.sp. *secalis* (Roberge ex Desmaz), *Prs*) bound to the promoter sequence, the aerial parts of plants of line L318 infected with the pathogen (BR+) or mock-treated (K) were sampled. Plants were cultivated in 24-well trays filled with a mixture of peat and perlite (at a ratio of 1:1). In one segment (7 cm), eight plants were grown. The seed preparation method and plant cultivation conditions were the same as described above. The methodology of inoculation was the same as described by Dmochowska-Boguta [17]. Leaf samples (infected or mock-treated) were collected from 12-day-old seedlings, 8 h after inoculation. The time from infection to sampling was established experimentally. The experiment was performed in three biological replicates, with two pots per replicate. The analysis performed on untreated plants included in the “brown rust” experiment was aimed at subtracting stress-specific proteins and identifying proteins specific for growth and development bound to the promoter sequences.

To identify promoter-bound proteins, 100 mg of the aerial parts of rye plants were ground in liquid nitrogen following extraction of the nuclei of cells using CelLytic™ PN Isolation/Extraction Kit (Sigma, St. Louis, MO, USA).

Promoter sequences were amplified by PCR (Mastercycler^®^ Nexus Gradient; Eppendorf, Hamburg, Germany) using the following program: 94 °C for 5 min; 40 cycles of three-step amplification of 30 s at 94 °C, 30 s at 60 °C, and 45 s at 72 °C; and then 5 min at 72 °C. Primers were designed based on bioinformatic analysis of 1000-nt upstream regions of both genes using the open access PlantCARE database [18]. Fragments with a high frequency of potential stress-specific motifs (SSMs) and growth- and development-specific motifs (GDSMs) were used as the templates for primer design (primer sequences are given in Table 2).

Amplified PCR products (485 and 661 nt for *ScBx1* and *ScIgl*, respectively) were purified according to the instructions of the PCR/DNA Clean-Up Purification Kit (Eurx) and eluted in 50 μL of elution buffer. The promoter sequence was used to capture proteins according to a modified version of a previously reported protocol [19]. Briefly, DNA was diluted to 20 pmol in 100 μL of PBS_50_ (10 mM PO_4_ at pH 7.4, 50 mM NaCl). Then, 20 μL of Dynabeads M-280 Streptavidin (Thermo Fisher Scientific) was washed with 200 μL of PBS_50_ followed by 1 h of incubation with DNA on a rotary wheel Heidolph REAX 2 (Heidolph Instruments, Schwabach, Germany) at 21 °C and then washed again three times with PBS_50_. The protein extract was mixed with 1.5 vol. of binding buffer (4 mM Hepes at pH 7.5, 120 mM KCl, 8% glycerol, 2 μM dithiothreitol, 0.166 μg/μL salmon sperm DNA, 0.166 μg/μL PolydIdC LightShift Poly dI-dC) and incubated with DNA on beads on a rotary wheel for 1 h at 21 °C, washed with 500 μL of binding buffer, and then washed again three times with 1 mL of PBS_50_ with 0.1% Tween 20. To free bound protein/DNA complexes from the beads, the samples were digested with fast digest *Eco*RI (Thermo Fisher Scientific) for 15 min at 37 °C. The proteins were analyzed on an LTQ-Orbitrap Elite (Thermo Fisher Scientific) mass spectrometer coupled with a NanoACQUITY LC system (Waters, Milford, MA, USA).

### 2.7. Biochemical Analysis

Quantitative analyses of BX, namely, HBOA, DIBOA, GDIMBOA, DIMBOA, and MBOA content, at 14, 21, 28, 42, 70, and 77 dag were carried out using a modified version of a previously published protocol [16]. Briefly, samples of plant material were mixed with diatomaceous earth and extracted with 70% methanol containing an internal standard, 2 µg/mL indoxyl β-D-glucoside (IbG), in stainless steel extraction cells of an accelerated solvent extraction system (ASE 200; Dionex, Sunnyvale, CA, USA). Extractions were carried out at an operating pressure of 10 MPa and 40 °C. Extracts were then evaporated to dryness under reduced pressure, reconstituted in 1 mL of 70% methanol containing 0.1% (*v/v*) acetic acid, and stored at −20 °C. Before the analyses, extracts were centrifuged for 20 min at 23,000× *g* and 4 °C and filtered with regenerated cellulose membrane filters with a pore size of 0.2 µm.

The analyses were performed using a Waters Acquity UPLC system (Waters) equipped with a triple quadrupole mass spectrometer (Waters TQD). Benzoxazinoids were separated on a Waters BEH C18 column (100 × 2.1 mm, 1.7 µm) with a linear, 8.5-min-long gradient from 3% to 15% acetonitrile containing 0.1% (*v/v*) formic acid (solvent B) in 0.1% formic acid (solvent A). All separations were carried out at 50 °C with a flow rate of 0.6 mL/min. After each gradient elution, the column was washed with 95% solvent B for 3 min and then re-equilibrated with 3% solvent B in solvent A for 3 min before the next injection. A 2.5 µL aliquot from each sample was injected using the “partial loop needle overfill” mode of a Waters Acquity autosampler.

The column effluent was introduced into the ion source of the mass spectrometer, which was operated in negative ion mode with the following parameters: capillary voltage −2.8 kV, extractor 3 V, RF lens 100 mV, source temperature 130 °C, desolvation temperature 400 °C, desolvation gas flow 1000 L/h, and cone gas flow 100 L/h. Collision cell entrance and exit were set to −2 and 0.5, respectively. Parameters of quadrupoles 1 and 3 were set to achieve unit-mass resolution. Cone voltage and collision energy were optimized for each compound to attain maximal response (Appendix A, also showing the details of the monitored ions). Data acquisition and processing were performed with Waters MassLynx 4.1 SCN 919 software.

Calibration curves ranging from 0.3 to 35 µg/mL were made by the appropriate dilution of DIBOA, DIMBOA, HBOA, GDIMBOA, MBOA, and IbG standard solutions (1 mg/mL each). GDIMBOA was used as a reference standard for GDIBOA quantitation. The concentration–response relationships for all investigated compounds were linear up to nearly 35 ng/µL. Samples in which concentrations of the measured compounds were higher than 30 ng/µL were appropriately diluted with 0.1% acetic acid and re-analyzed.

### 2.8. Statistical Analysis

Using Statgraphics Plus ver. 3.0 software, Kruskal-Wallis and Fisher’s LSD tests were performed to evaluate the significance of differences (at α = 0.05) between gene expression and BX synthesis levels at a given time point. Some calculations were carried out with the use of the R package [20] and with the use of the agricolae package [21].

## 3. Results

### 3.1. The Expression Profiles of the ScBx1 and ScIgl Genes

*ScBx1*: The expression level of *ScBx1* increased from the 14th to the 21st dag in all inbred lines, with the greatest increase (nearly 4.5-fold) in line L318. Next, on the 28th dag, its level decreased in two lines, L318 and D33, while in line D39 the transcripts were no longer detectable. At the fourth time point (42nd dag), the expression of *ScBx1* was measurable only in line D33. Starting from 70th dag, no expression of the *ScBx1* gene was detected. The most notable change was observed between the first and second time points. All lines were found to have very similar developmental *ScBx1* expression profiles, that is, an increase at the second time point and then a decrease on the 28th dag; however, they differed in terms of the level and the duration of detectability, with the highest values being recorded for line D33. Nevertheless, the differences between the expression level of the *ScBx1* gene at subsequent time points within each of the three tested lines were statistically not significant (Figure 2).

*ScIgl*: Starting from the first tested time point (14th dag), it was possible to detect the expression of the *ScIgl* gene in all three lines, albeit at a very low level. At the next time point (21st dag), its expression increased in line D39 (over 2.6-fold; being only slightly lower, with no statistically significant difference, than the expression level of *ScBx1*), while it remained at almost the same level in line D33 and decreased (more than 4-fold) in line L318. After the next 7 days, expression of the *ScIgl* gene increased in lines L318 and D33, while in D39 it fell. On the 42nd dag, the expression level of *ScIgl* dropped and on the 70th it increased in all tested lines. At the last time point (77th dag), the expression of *ScIgl* was detectable only in lines D33 and D39. Up to the 70th dag, line D39 was characterized by the highest expression level of the *ScIgl* gene, when line L318 showed its lowest level. Each inbred line was characterized by a unique *ScIgl* expression developmental pattern. The highest similarities were found for lines L318 and D33, which were found to have relatively similar expression profiles between the 21st and 70th dag. The greatest differences were observed at the second time point, when the expression level of *ScIgl* in line D39 was 152 and 32 times higher than those in lines L318 and D33, respectively (Figure 3).

*ScBx1* vs. *ScIgl*: After the first 14 days of cultivation, the expression level of the *ScBx1* gene was much higher than that of the *ScIgl* gene in each inbred lines, almost 12-, 64-, and 2-fold in L318, D33, and D39 respectively. At the next two time points, this relationship remained the same in lines L318 and D33. However, in line D39, *ScBx1* and *ScIgl* were expressed at almost the same level on the 21st dag (with no statistically significant differences recorded). Moreover, at the next time point (28th dag), the expression of *ScIgl* was much higher than that of *ScBx1*. On the 42nd day, the expression of *ScBx1* was detected only in line D33. At this time point, *ScIgl* was the only gene expressed in the two remaining lines (L318 and D39). Starting from the 70th dag, *ScIgl* was the only gene with measurable expression in all inbred lines, with the highest level in line D39 at the 70th dag and in line D33 at the 77th dag (Figure 4).

### 3.2. The Content of BX in Aboveground Parts of Rye Inbred Lines L318, D33, and D39

The chemical analyses showed that two compounds, DIBOA and GDIBOA, dominated the BXs in all lines and at all-time points (Figure 5A–C). The highest concentration of DIBOA was detected on the 14th dag in all tested lines, while the highest concentration of GDIBOA was on the 14th dag in lines L318 and D39, and on the 21st dag in line D33. The content of the other analyzed BXs was very low (particularly in line D39), but still detectable. The developmental patterns of BXs, especially DIBOA and GDIBOA, were similar in lines L318 and D39 up to the 42nd dag. In general, their contents decreased continuously in this period, except for DIBOA on the 28th dag and GDIBOA on the 21st dag in line D33. Starting from the 42nd dag, the contents of DIBOA and GDIBOA in lines L318 and D33, and GDIBOA in line D39 were still falling, while DIBOA in line D39 had begun to increase at the last time point. In the other lines, considerable increases in the content of both BXs was observed at the last time point. Between the 70th and 77th dag, the content of the remaining BXs (except for DIMBOA) increased; in the case of MBOA, this increase was statistically significant (in lines D33 and D39).

### 3.3. Correlations between Gene Expression Level and BX Content

In all lines, the profile of BX synthesis usually corresponded with the profile of *ScBx1* gene expression up to the 28th dag, with the exception of the second time point (21st dag), when its expression level increased but content of the majority of BXs decreased (Figure 2 and Figure 5A–C). The same reaction was observed for *ScIgl* gene in line D39.At the subsequent time points, the contents of BXs were usually more in line with the expression level of the *ScIgl* gene (Figure 3 and Figure 5A–C). Spearman’s rho statistics showed, however, that the majority (over 70%) of correlations between the expression levels of both genes and the content of the two dominant BXs were not statistically significant (Table 3; the values of r coefficients calculated based on Spearman’s rho are included in Appendix A).

### 3.4. In Silico and in vivo Characteristics of 1000 nt 5′ Upstream Regulatory Sequences of the Genes ScBx1 and ScIgl

#### 3.4.1. *In Silico* Promoter Analysis

##### Stress-Specific Motifs

In the 1000 nt promoter sequences of the *ScBx1* and *ScIgl* genes, the two same types of SSMs, namely, a cis-acting element involved in abscisic acid responsiveness and a cis-acting regulatory element involved in MeJA responsiveness, were recognized. Two more types of SSMs, namely, MYBHv1 binding site and MYB binding site involved in drought inducibility, were specific for the *ScBx1* and *ScIgl* genes, respectively. The overall frequencies of SSMs in the promoters of both genes differed significantly: In the promoter of the *ScIgl* gene, this frequency was over three times higher than in the promoter of the *ScBx1* gene. The highest SSMs frequency was found for the cis-acting element involved in abscisic acid responsiveness identified in the *ScIgl* gene (Table 4).

##### Growth- and Development-Specific Motifs

Bioinformatic analysis of the 1000 nt promoter sequence allowed us to detect several motifs that could be related to growth and developmental processes; specifically, there were one and six of these in the promoters of the *ScBx1* and *ScIgl* genes, respectively. The first motif, auxin-responsive element, was common for both genes, but in the promoter of the *ScIgl* gene it was five times more common. The second motif, cis-acting element involved in gibberellin responsiveness, was found only in the promoter of the *ScIgl* gene (Table 4).

#### 3.4.2. *In Vivo* Promoter Analysis

Based on the bioinformatic analysis of 1000 nt 5′ upstream regions of both genes, the fragments of 485 nt and 660 nt in length comprising all in silico-identified motifs (SSMs and GDSMs) of promoter sequences of the *ScBx1* and *ScIgl* genes were included in the in vivo assay.

In *Prs*-treated plants, overall, 20 transcription factors (TFs) attached to the promoter of the *ScBx1* gene, while 9 did so to the promoter of the *ScIgl* gene; three TFs were common to both genes. In unstressed plants, 14 and 6 regulatory proteins were found to bind to the promoters of the *ScBx1* and *ScIgl* genes, respectively; 4 of them were common to both genes (Appendix A). From these pools, 7 and 5 TFs for *ScBx1* and *ScIg*l, respectively, were annotated in terms of their potential role in stresses and processes of growth and development (Table 5, Appendix A).

##### Stress-Specific (SS) Transcription Factors

Among all annotated TFs identified in the protein pool isolated from *Prs*-treated plants, four were somewhat associated with stress response: four proteins bound to the promoter of the *ScBx1* gene and one bound to the promoter of the *ScIgl* gene. The germin-like protein linked to the promoter of the *ScIgl* gene was also found among the TFs bound to the promoter of *ScBx1*. In addition to regulatory proteins associated with stress, one TF specific for growth and development (zinc finger protein ZAT1_1) was also identified in these protein pool (Table 5, Appendix A).

##### Growth- and Development-Specific (GDS) Transcription Factors

Overall, five TFs identified in the protein pool isolated from untreated plants that can be assigned roles in regulating growth and development processes have been identified. Three proteins were bound to each of the promoters (Table 5). Among them, one TF, namely, RING-H2 finger protein, was common to both genes (Table 5, Appendix A).

### 3.5. Expression Profile of the ScIgl Gene in Rye Plants with Virus-Induced Silenced ScBx1 Gene

The expression of the *ScIgl* gene was detected in all selected plants with considerably or completely silenced expression of the *ScBx1* gene. Higher levels of normalized expression were detected in plants at 14 dpi (mean 2.00 ± 2.43, median 1.23) than in plants at 21 dpi (mean 0.67 ± 0.59, median 0.32). Plants #9 and #13 were characterized by the highest normalized expression level of the *ScIgl* gene, at 14 and 21 dpi, respectively. In the plants without complete silencing of the *ScBx1* gene, the expression of *ScIgl* was 3.5 (plant #5) to 313 (plant #9) and 6.4 (plant #10) to 37 (plant #8) times higher than *ScBx1* at 14 dpi and 21 dpi, respectively (Figure 6 and Figure 7, Table 6, Appendix A).

## 4. Discussion

It has generally been considered that the *Bx1* gene (and its orthologs) is under developmental control and is associated with BX production under native conditions, whereas the *Igl* gene is inducible by stress signals, such as wounding, herbivory, or jasmonates [13,22]. The results reported by Groszyk et al. [14], however, contradict this. First, in this previous study, the authors showed that the *ScBx1* gene was expressed in unstressed rye plants over the course of development and, on the 21st day, it became undetectable. In contrast, after virus infection it increased more than 40-fold. This clearly indicates that the *ScBx1* gene is regulated both developmentally and by stress. Second, they found that, starting from the third week of cultivation, the *ScBx1* gene was not expressed any more while BXs (HBOA, DIBOA, DIMBOA, GDIMBOA, and MBOA) were still synthesized at almost constant levels up to the 99th day. In their article, the authors concluded that “based on the obtained results, we anticipate the presence of the *Igl* ortholog in rye, which contributes along with *ScBx1* to BX biosynthesis in rye. We expect that the gene will be constitutively expressed at least in plants older than 21 days”. Third, the plants with silenced *ScBx1* were proven by the authors to produce BX despite an almost complete absence of *ScBx1* transcripts. All of these findings prompted us to undertake the study presented in this paper. We formulated the hypothesis that, in older rye plants, the role of the *ScBx1* gene is taken over by another gene encoding IGL, most probably *ScIgl*, because the rye ortholog the *TSA* gene participating in another pathway, namely, tryptophan biosynthesis, had to be ruled out for obvious reasons.

### 4.1. Gene Expression: ScBx1 vs. ScIgl

Our results showed that the ratio between the expression levels of the two genes *ScBx1* and *ScIgl* changes over time. The expression of *ScBx1* decreased and eventually became undetectable about the 42nd dag in lines L318 and D39, and near the 70th dag in line D33. These findings are basically consistent with the results presented previously by Groszyk et al. [14]. In addition, Tanwir et al. [10] reported that the *ScBx1* gene was expressed at the highest level at the early developmental stages of 24–30 h after germination, followed by a decrease at later stages. In contrast to the *ScBx1* gene, the *ScIgl* gene was expressed in all lines up the 77th dag, despite the lack of any stressor.

### 4.2. Predicted Function of ScBx1 and ScIgl Based on Promoter Analysis

The in silico analysis of promoters provides preliminary information about gene function. Such an approach has been applied with respect to other genes potentially associated with defense against stresses, such as the rye *ScBx* genes, including *ScBx1* [9], *Arabidopsis thaliana* pathogenesis-related *PR* genes [23], and rice *OsGLP* genes encoding germin-like proteins [24]. In the present study, four stress-specific motifs were identified bioinformatically. Two of them, namely, cis-acting element involved in ABA responsiveness and cis-acting regulatory element involved in methyl jasmonate (MeJA) responsiveness, are present in the promoters of both genes, but their general frequencies differ markedly, being much higher in the case of the promoter sequence of the *ScIgl* gene. ABA is a plant stress hormone related to abiotic stresses, such as cold, salinity, or drought [25,26]. Thus, ABA-responsive elements, sites for binding specific transcription factors, are common in stress-responsive genes [27,28]. MeJA is a signaling molecule that regulates plant responses to biotic stresses, such as wounding caused by insects and infection by fungal pathogens, as well as to abiotic stresses, such as drought [29]. Two other motifs, MYBHv1 and MYB binding site, involved in drought inducibility were unique to either *ScBx1* or *ScIgl*, respectively. Nevertheless, both motifs belong to the same family of transcription factors (MYB), which is mainly involved in ABA-dependent stress signaling pathways [26,30]. It can therefore be assumed that both *ScBx1* and *ScIgl* are stress-responsive genes, but the much higher frequency of SSMs in the promoter of *ScIgl* might suggest that this gene plays a more significant role in stress response reactions than *ScBx1*; this generally matches current opinion. However, the results of parallel studies conducted at our department indicate that *ScIgl* tends to be downregulated by stresses such as infection with brown rust and co-cultivation with Berseem clover, but upregulated by a low temperature during vernalization [8]. Moreover, from the results of bioinformatic analysis aimed at finding GDSMs, it could be concluded that the *ScIgl* gene plays a greater role in the processes of growth and development than the *ScBx1* gene, as the frequency of these motifs (auxin-responsive element and cis-acting element involved in gibberellin responsiveness) was six-fold higher in *ScIgl* than in *ScBx1*. Furthermore, the second one mentioned above motif was found only in the promoter of the *ScIgl* gene.

The in silico analysis of promoter sequences of both genes was performed not only to preliminarily verify their function, but also to select fragments with high frequencies of SSMs and GDSMs for a wet experiment. In the in vivo assay, 49 proteins were found to bind to the *ScBx1* and *ScIgl* promoter sequences selected based on bioinformatic analysis; 4 of them can be considered as stress-specific and 5 as growth- and development-specific. Among the TFs identified under stress conditions, only one (germin-like protein) was common to both genes. GLPs are encoded by a family of genes found in all plants and are associated with responses to biotic (viruses, bacteria, mycorrhizae, fungi, insects, nematodes, and parasitic plants) and abiotic stresses (salt, heat/cold, drought, nutrient, and metal), and especially with fungal pathogenesis in cereals [24,31]. The binding of this stress-specific TF to the promoters of *ScBx1* and *ScIgl* suggests that both genes are regulated by stress. However, the binding to the promoter of *ScBx1* of two other proteins (NIM1-interacting TFIIH subunit and myb-like DNA-binding domain), each of which has been proven to be associated with biotic and/or abiotic stresses [32,33,34,35], may indicate the greater roles of these genes in defense reactions. In the case of unstressed plants, one common TF (RING-H2 finger protein ATL8-like) bound to the promoter sequences of both genes. This regulatory protein plays an important role in the adaptation of plants to abiotic stresses [36,37]. In previous studies, no findings of a correlation of the TF RING-H2 finger protein ATL8-like with these processes were reported. Our results may indicate its novel function, but this needs to be confirmed through further experiments.

Similarly, as in plants inoculated with brown rust, unique TFs for the gene *ScBx1*, agamous-like (AGL) and EXORDIUM-like 2 (EXOL2), were identified in untreated plants. AGL, as a member of the MADS-box gene family, plays an important role in the regulation of plant growth and development [38]; EXOL2 has been shown to play a role in meristem function [39] and to suppress brassinosteroid-dependent growth [40,41]. In turn, we identified one specific gds TF as being linked only with the promoter of the second gene, namely, NAC domain-containing transcription factor, which acts as a master regulator of xylem vessel differentiation [42]. Taking these results into account, we can conclude that both genes are developmentally regulated, but the findings for the *ScBx1* gene are more robust.

In summary, on the basis of in silico and in vivo analyses of 5′ regulatory sequences, we cannot say unequivocally that any of the two genes is regulated in a certain way. Instead, the results of these analyzes and other experiments described here indicate that both genes are regulated developmentally and by stresses, but they are expressed at different developmental stages. This contradicts the current consensus that *ScBx1* is regulated only developmentally, while *ScIgl* is regulated only by stress.

### 4.3. Whether and How the Expression of ScBx1 and ScIgl Genes Is Related to BX Synthesis

The next issue that we aimed to clarify was the relationship between the expression profiles of both genes and BX synthesis. Among the BXs, two compounds dominated: DIBOA and GDIBOA. Groszyk et al. [14] also found DIBOA to be a dominant BX. The authors observed, however, a gradual increase of its content up to the 14th day, then the drastic decrease followed by continuous, slow decline until the end of the experiment, at the 99th day. The content of GDIBOA in their experiments decreased continuously but it was still detectable at the 99th day. The differences in developmental profiles of DIBOA and GDIBOA observed by Groszyk et al. [14] and in our study may have resulted from genotypic determinants. Groszyk et al. [14] used hybrid variety Stach F_1_, which is not related to the inbred lines investigated in our work. Unfortunately, it is not possible to compare our results with other reports on rye because, with the exception of Groszyk et al. [14], no other researchers have analyzed the content of BXs in such old plants as ours.

We observed that the content of BXs gradually decreased up to 70th dag and in the last week, it increased (except for line D39, in which DIBOA content began to increase from the 42nd dag). Starting from the 42nd, 70th, and 28th dag, the expression of the *ScBx1* gene became undetectable in lines L318, D33, and D39, respectively. In contrast, the expression of the *ScIgl* gene, although at a low level, was still detectable. This observation leads to the conclusion that the *ScIgl* gene takes over the role of *ScBx1* at later developmental stages and controls the first step of BX biosynthesis, requiring the production of DIBOA and GDIBOA. The correlations between the levels of expression of both genes, *ScBx1* (at earlier stages) and *ScIgl* (at later stages), and BX synthesis were not always significant, but the relationship is nonetheless clear.

### 4.4. The Inbred Lines Differed in Respect to Gene Expression Level and Pattern, and BX Content

Despite the similar tendencies being found in all inbred lines, they differed with respect to the expression levels of both genes, *ScBx1* and *ScIgl*, as well as the BX content and their developmental profiles. Line D33 was characterized by the highest expression of *ScBx1* and line D39 showed the highest expression of *ScIgl*, particularly at the second time point. D33 was the only line in which the *ScIgl* expression level increased gradually between the 42nd and 77th dag. These dissimilarities can result from the differences in the 5′ regulatory sequences between these inbred lines, among other factors. In our previous study, we detected several polymorphisms in the *ScBx1* gene associated with the content of BX. At two polymorphic sites, the first one positioned in the gene promoter line and the second one in 3′ UTR, lines D33 and D39 possessed the same, “positive” nucleotides, namely, T and G, while line L318 had the “negative” nucleotides G and T, respectively [43].

The highest content of BXs was found in line D33, while the lowest one was in line D39. The same relationship was observed in the case of field-grown plants with respect to DIBOA, but not for GDIBOA; line L318 was characterized by the highest content of this BX [43].

In all lines, the content of BXs generally corresponded with the expression levels of the *ScBx1* (at earlier stages) and *ScIgl* genes (at later stages). Only a few cases of deviation from this relationship, namely, a considerable increase in gene expression and a decrease in BX synthesis (and also vice versa) were observed, specifically, on the 21st dag for GDIBOA and DIBOA in lines L318 and D39, on the 28th dag for DIBOA in line D33, on the 77th dag for DIBOA in lines L318 and D39, and on the 77th dag for GDIBOA in line L318.

Overall, lines L318 and D39 showed the greatest similarity in terms of gene expression and BX profiles. This is in accordance with the values of the Dice similarity coefficient calculated based on data obtained previously [16,43], which were 0.508, 0.523, and 0.546 for pairs D33/D39, D33/L318, and D39/L318, respectively.

### 4.5. ScIgl Is Expressed in Plants with Silenced ScBx1

The final confirmation of the hypothesis that the *ScIgl* gene may take over the role of *ScBx1* was obtained from the experiment using cDNA of plants that silenced the *ScBx1* gene. The analyzed plants were proven by the authors to produce BX despite the almost complete absence of *ScBx1* transcripts, especially at 14 dpi [14]. Although the level of BX in plants with silenced *ScBx1* was lowered (3× and 3.9× lower, respectively, at 14 and 21 dpi) the reduction of *ScBx1* gene expression was as high as 62.5× and 4× compared to the control (plants treated with empty vector). In this context, the following question arises: Which of the genes supplies free indole to the BX biosynthesis pathway? The results presented in this paper clearly indicate that it is the *ScIgl* gene that acts in this way and confirm earlier suggestion of Groszyk et al. [14] about this gene.

Our observations discussed above as well as those reported by Groszyk et al. [14] enable us to assert that both *ScBx1* and *ScIgl* genes are not only regulated developmentally but are also involved in defense responses. However, the first gene is expressed at earlier stages and the second one at later stages. When the expression of *ScBx1* decreases, the expression of *ScIgl* increases over time, although its level is low. Consequently, the *ScBx1* gene provides indole to the BX biosynthesis pathway at earlier developmental stages and *ScIgl* at later ones (Table 7).

The obtained results showed new aspects of the genetic background of benzoxazinoid biosynthesis and proved that its first stage can be regulated in an alternative way which should lead to a revision of existing views on this subject.

## Figures and Tables

**Figure 1 genes-11-00223-f001:**
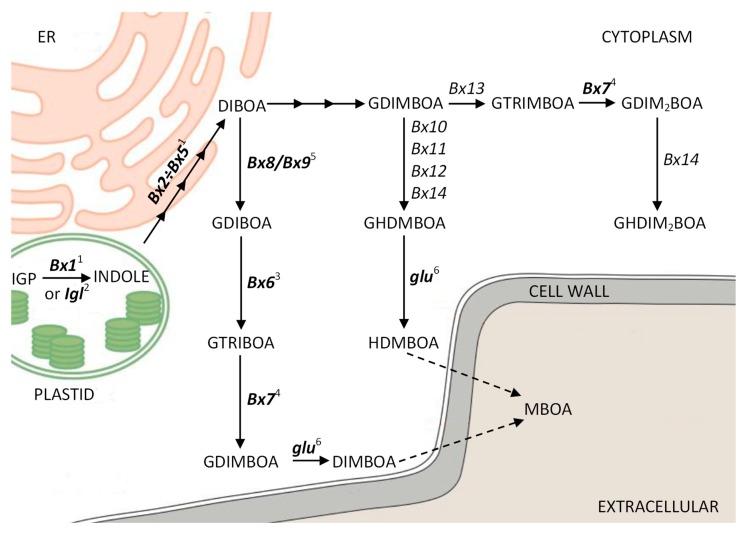
Biosynthesis pathway of benzoxazinoids in maize (*Zea mays*) (modified scheme according to Niculaes et al. [15]). Rye orthologs of *Bx* genes isolated and sequenced are marked in bold, ^1^ KF636825–KF636828 and KF620524, ^2^ MN120476, ^3^ HG380520, ^4^ MG519859, ^5^ AB548283.1, ^6^ AY586531.2.

**Figure 2 genes-11-00223-f002:**
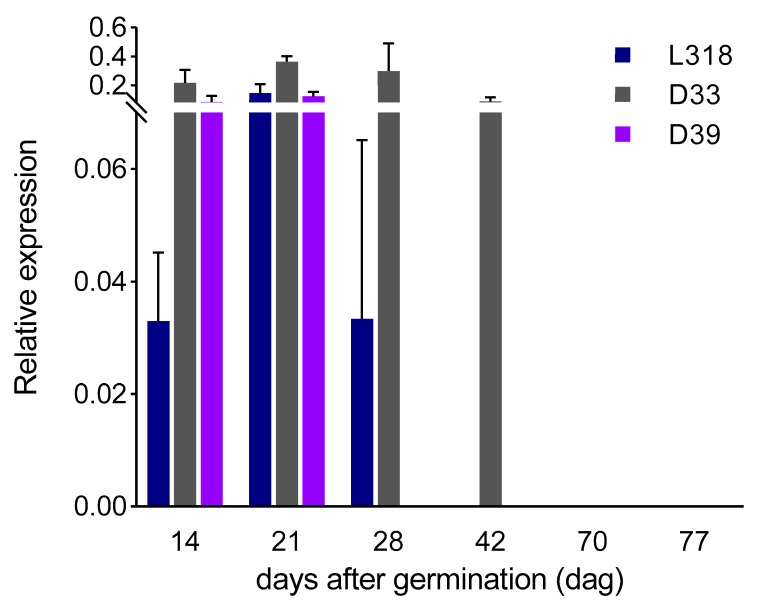
Expression patterns of the *ScBx1* gene of three rye inbred lines, L318, D33, and D39 at six time points, 14, 21, 28, 42, 70, and 77 dag. The data represent mean value with standard deviation. There is no statistically significant difference between the expression level of the *ScBx1* gene at subsequent time points within each of the three tested lines.

**Figure 3 genes-11-00223-f003:**
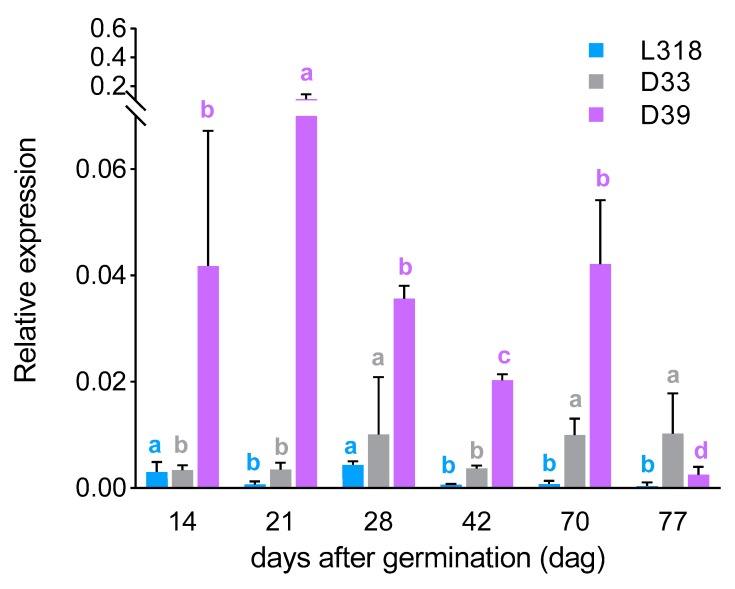
Expression patterns of the *ScIgl* gene of three rye inbred lines, L318, D33, and D39 at six time points, 14, 21, 28, 42, 70, and 77 dag. The data represent mean value with standard deviation. The letters a, b, c, and d denote statistically significant (*p* ≤ 0.05) differences in relative expression level between subsequent time points in a given line.

**Figure 4 genes-11-00223-f004:**
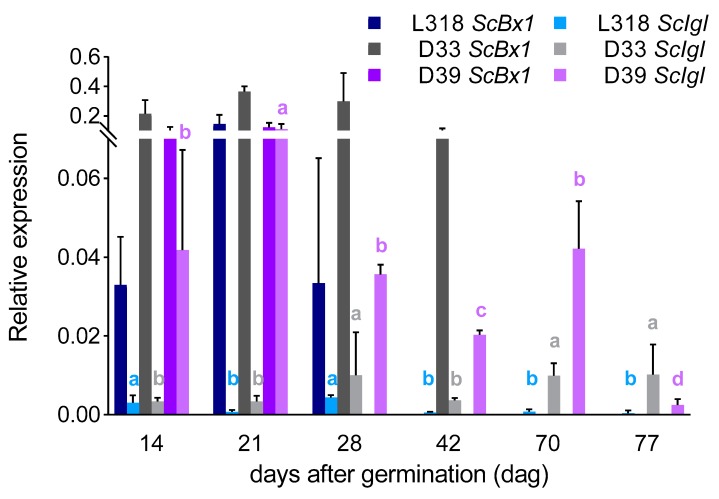
Comparison of the expression patterns of the *ScBx1* and *ScIgl* genes of three rye inbred lines, L318, D33, and D39 at six time points, 14, 21, 28, 42, 70, and 77 dag. The data represent mean value with standard deviation. The letters a, b, c, and d denote statistically significant (*p* ≤ 0.05) differences in relative expression of the studied genes between successive time points within one line.

**Figure 5 genes-11-00223-f005:**
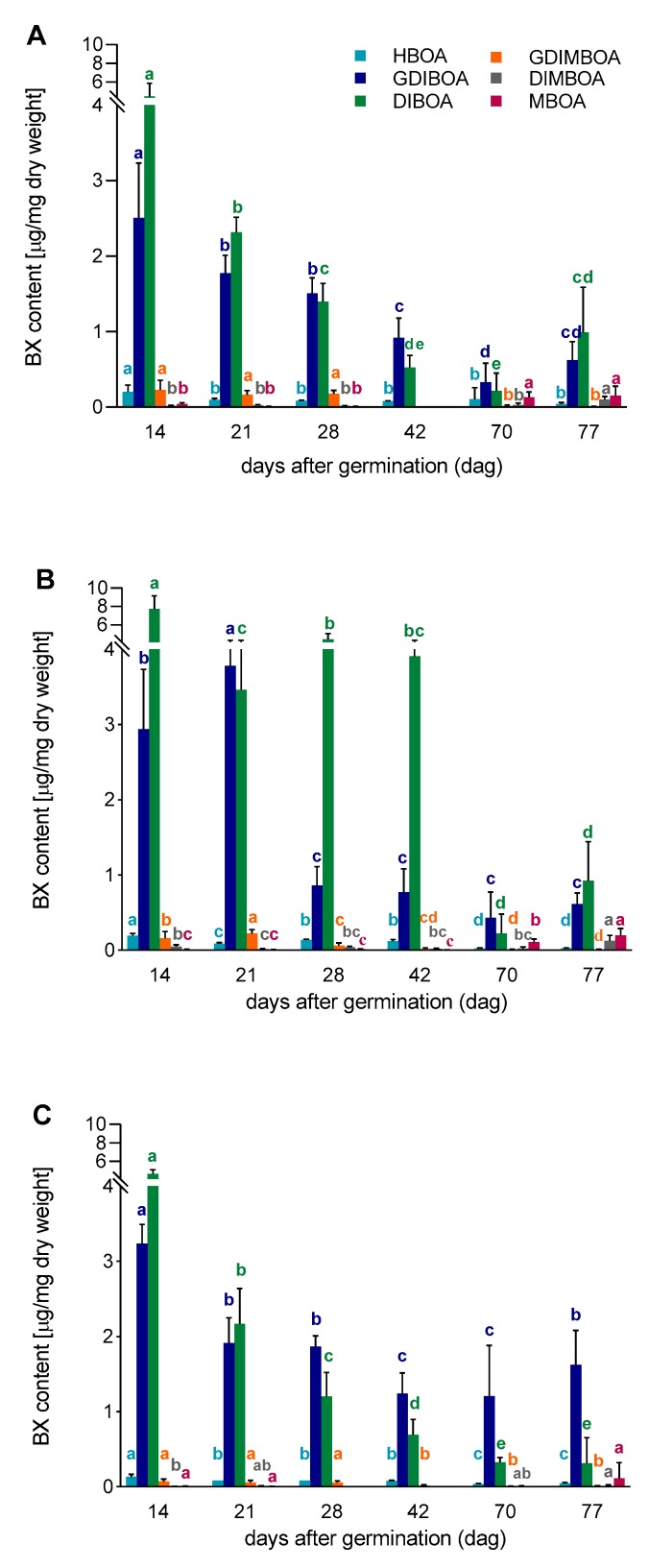
Benzoxazinoid (BX) content in aboveground parts of rye lines, (**A**) L318, (**B**) D33, (**C**) D39 at six time points, 14, 21, 28, 42, 70, and 77 dag. The data represent mean value with standard deviation. The letters a, b, c, d and e denote statistically significant (*p* ≤ 0.05) differences between subsequent time points in a given BX.

**Figure 6 genes-11-00223-f006:**
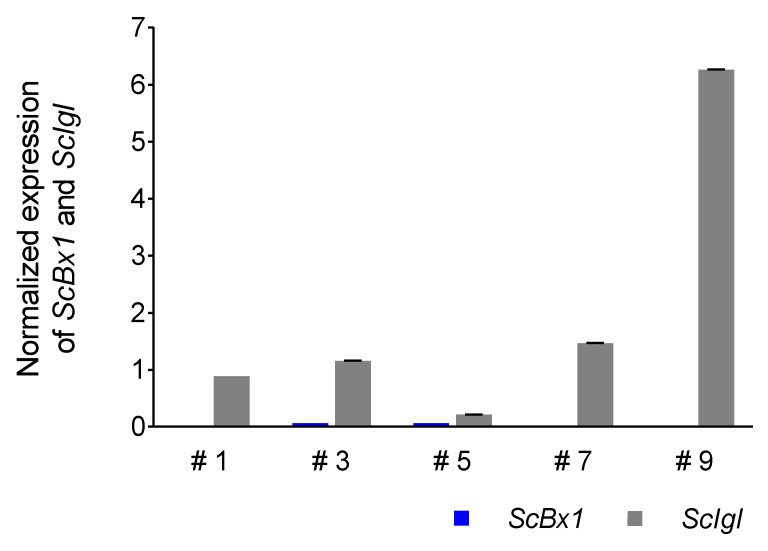
Patterns of normalized (in respect to the empty BSMV:*PDS* vector) expression of *ScIgl* and *ScBx1* genes in leaves of rye cv. Stach F_1_ inoculated with BSMV:*ScBx1* on the 14th dpi, for methodological details related to silencing procedure see Groszyk et al. [14]. The results represent mean value and standard deviations of three technical replicates.

**Figure 7 genes-11-00223-f007:**
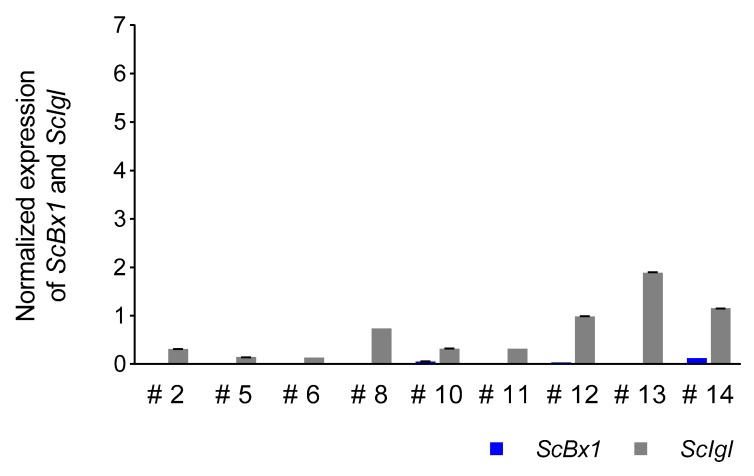
Patterns of normalized (in respect to the empty BSMV:*PDS* vector) expression of *ScIgl* and *ScBx1* genes in leaves of rye cv. Stach F_1_ inoculated with BSMV:*ScBx1* on the 21st dpi; for methodological details related to silencing procedure see Groszyk et al. [14]. The results represent mean value and standard deviations of three technical replicates.

**Table 1 genes-11-00223-t001:** Primers used in qRT-PCR reactions.

Gene	Sequences (5′–3′)
*ScBx1*	F: TCAAAACCTGAACACGTGAAGC
	R: GCCTCTAGCCTTTTCAATCCTTC
*ScIgl*	F: AACACCAGCTACACCATCAGAG
	R: GTGGGTTTACAGTCGCCCTA
*HvAct*	F: CCCCTTTGAACCCAAAAGCC
	R: GAAAGCACGGCCTGAATAGC
*ScGADPH*	F: GAGTCTGCCCACCCATTCGTAA
	R: GACATGCCATAGGTTTCAGCGAC
*Ta54227*	F: CAAATACGCCATCAGGGAGAACATC
	R: CGCTGCCGAAACCACGAGAC

**Table 2 genes-11-00223-t002:** Primers used in PCR reactions.

Gene	Sequences (5′–3′)
*ScBx1*	F: **CCGGAATTCCGG**AAATTTGCCCGGTCTACGTG
	R: CGCATACAACCAAACACCAGC
*ScIgl*	F: **CCGGAATTCCGG**GCGTCCATCTTCACGTTGAT
	R: GCTGTGTTGGCAGGTAGTGT

*Eco*RI cleavage site and neighboring nucleotides are marked in bold.

**Table 3 genes-11-00223-t003:** Correlations between the expression levels of the genes *ScBx1* and *ScIgl*, and the content of GDIBOA and DIBOA at six subsequent time points.

Line	Gene	GDIBOA						DIBOA					
		Time Point [dag]						Time Point [dag]					
		14	21	28	42	70	77	14	21	28	42	70	77
L318	*ScBx1*	(-) *	(+)	(-)	nd	nd	nd	(+)	(+)^*^	(-) **	nd	nd	nd
	*ScIgl*	(+)	(+)	(+)	(+)	(+)	(-)	(-)	(-) **	(+)	(+)	(+)	(+)
D33	*ScBx1*	(+)	(-)	(-) **	(-) **	nd	nd	(+)	(-) **	(-)	(-)	nd	nd
	*ScIgl*	(-)	(-)	(+)	(+) **	(+)	(-)	(-)^*^	(-)	(+)	(+)	(+)	(-)
D39	*ScBx1*	(-)	(+) **	nd	nd	nd	nd	(-)	(-) **	nd	nd	nd	nd
	*ScIgl*	(-) **	(-)	(+)	(-) **	(+)	(-)	(-) **	(+)	(+)	(-)	(+) **	(+)

Positive correlations: (+), gray in color; negative correlations: (-); nd: the expression of *ScBx1* gene no longer detectable; *: significant at *p* < 0.1; **: significant at *p* < 0.05.

**Table 4 genes-11-00223-t004:** Stress-specific and growth- and development-specific motifs in promoter sequences of the *ScBx1* and *ScIgl* genes.

Promoter of the Gene:	Motif Type	Motif Sequence/No. of a Given Motif	Probable Function	Frequency [No. of SSMs (or GDSM)/100 nt]
*ScBx1*	SSM	ACGTG/1	cis-acting element involved in abscisic acid (ABA) responsiveness	0.1
		CCGTTG/1	**MYBHv1 binding site**	0.1
		TGACG/1	cis-acting regulatory element involved in MeJA-responsiveness	0.1
		Mean		0.3
*ScIgl*		CACGT/2	cis-acting element involved in abscisic acid (ABA) responsiveness	0.5
		ACGTG/1		
		TGACG/2		
		CAACTG/1	**MYB binding site involved in drought-inducibility**	0.1
		TGACG/2	cis-acting regulatory element involved in MeJA-responsiveness	0.4
		ACGAC/1		
		CGTCA/1		
		Mean		1.0
*ScBx1*	GDSM	GTCGTT/1	auxin-responsive element	0.1
		Mean		0.1
*ScIgl*		TATCCA/1	**cis-acting element involved in gibberellin responsiveness**	0.1
		GTCGTT/1	auxin-responsive element	0.5
		AACGAC/2		
		CGACGA/1		
		GTCTT/1		
		Mean		0.6

Motifs unique for a given gene are marked in bold.

**Table 5 genes-11-00223-t005:** Regulatory proteins related to stress response and growth and development bound to the promoters of the *ScBx1* and *ScIgl* genes.

Promoter of the Gene:	TF Type	Protein Name (Acc. No.)	Function	Frequency [No. of Bound Proteins/100 nt]
*ScBx1*	ss	germin-like protein (AEN02469.1)	response of plants to biotic (viruses, bacteria, mycorrhizae, fungi, insects, nematodes, and parasitic plants) and abiotic stresses (salt, heat/cold, drought, nutrient, and metal)	
		**NIM1-interacting TFIIH subunit (POO02874.1)**	potentiates plant disease resistance and results in enhanced effectiveness of fungicides; key regulator of systemic acquired resistance in plants	
		**myb-like DNA-binding domain (XP_026398713.1)**	regulates various cellular processes, including cell cycle and cell morphogenesis, biotic and abiotic stress responses	
		**zinc finger protein ZAT1_1 (PWZ38869.1)**	plays key roles during plant growth and development, and a number of zinc finger TFs were shown to be involved in plant abiotic and biotic stresses	
				0.82
*ScIgl*		germin-like protein (AEN02469.1)	response of plants to biotic (viruses, bacteria, mycorrhizae, fungi, insects, nematodes, and parasitic plants) and abiotic stresses (salt, heat/cold, drought, nutrient, and metal)	
				0.15
*ScBx1*	gds	RING-H2 finger protein ATL8-like (XP_020242355.1)	an important role in plant adaptation to abiotic stresses	
		**protein EXORDIUM-like 2 (XP_002960721.1)**	a component in BR signaling-mediated (BR-promoted) growth; it is hypothesized that EXL1 suppresses brassinosteroid-dependent growth	
		**agamous-like MADS-box protein (XP_027364274.1)**	MADS-box gene family plays an important role in the regulation of plant growth and development and is well known as a key group of transcription factors	
				0.62
*ScIgl*		RING-H2 finger protein ATL8-like (XP_020242355.1)	an important role in plant adaptation to abiotic stresses	
		**NAC domain-containing transcription factor (NAC042); ANR02348.1**	acts as a master regulator of xylem vessel differentiation	
		**zinc finger protein ZAT1_1 (PWZ38869.1)**	plays key roles during plant growth and development, and a number of zinc finger TFs were shown to be involved in plant abiotic and biotic stresses	
				0.45

TFs unique for a given gene bound to promoters of both genes under the same conditions (stress or native) are marked in bold.

**Table 6 genes-11-00223-t006:** Multiplied normalized expression level of the *ScIgl* gene with respect to the *ScBx1* gene in plants with considerable silenced of latter’s expression.

Plant	Dpi	Ratio
#3	14	19.3
#5		3.5
#9		313
#6	21	13
#8		37
#10		6.4
#12		32.67
#14		9.58

Ratio means normalized expression level of *ScIgl*/normalized expression level of *ScBx1*.

**Table 7 genes-11-00223-t007:** The “turn on” gene in BX biosynthesis pathway.

Time Point [Dag]	IL		
	L318	D33	D39
14	*ScBx1*	*ScBx1*	*ScBx1*
21	*ScBx1*	*ScBx1*	*ScBx1*
28	*ScBx1*	*ScBx1*	*ScIgl*
42	*ScIgl*	*ScBx1*	*ScIgl*
70	*ScIgl*	*ScIgl*	*ScIgl*
77	*ScIgl*	*ScIgl*	*ScIgl*

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
