# Peer review of "Genes ScBx1 and ScIgl—Competitors or Cooperators?"

_genes, 2020, doi:10.3390/genes11020223_

Round 1
Reviewer 1 Report
I had an opportunity to review manuscript “Genes ScBx1 and ScIgl – Competitors or Cooperators?” for future publication in Genes. This manuscript by Wlazlo and coauthors describes that the indole biosynthesis in rye is controlled by the ScBx1 and ScIgl genes, which are both regulated developmentally and by estresses. Previous results from Groszyk et al (reference 14 in the manuscript) contradicted this; however, expression studies of ScBx1 and ScIgl, measures of BX content, and the in vivo and in silico promoter analysis carried out by Wlazlo and col. in this project confirmed the cooperative role of ScBx1 and ScIgl during development and in stress situations.
-Figure 1: The letters (stadistic analysis) of the line D39 are missing. Please, correct it.
- Line 227: “After the first 14 days of cultivation, the expression level of the ScBx1 gene was much higher than that of the ScIgl gene in all inbred lines”à I cannot see this affirmation in the line D39, in which I only can see an increase of two fold in the ScBx1 expression with respect to the ScIgl expression. Please, describe it adequately.
-Figure 3: The letters (stadistic analysis) of the line D39 ScBx1 are missing. Please, correct it.
- Line 262: “In all lines, the profile of BX synthesis usually corresponded with the profile of ScBx1 gene expression up to the 28th dag, with the exception of the second time point (21st dag), when its expression level increased but content of the majority of BXs decreased (Figure 1, 4A–C)”. à I think that this affirmation is not exact since the profile of BX synthesis also corresponded with the profile of ScIgl gene expression from line D39 up to the 28th dag, with the exception of the second time point (21st dag). Please describe it adequately.
Author Response
Point 1: Figure 1: The letters (stadistic analysis) of the line D39 are missing. Please, correct it.
Response 1: We have removed the letters from the Figure 2 (before Figure 1) as the Kruskal-Wallis test did not detected any differences between the values of expression levels at subsequent time points in any of inbred lines. The appropriate comment is given in the result description: “Nevertheless, the differences between the expression level of the ScBx1 gene at subsequent time points within each of the three tested lines were statistically not significant (Figure 2)”.
Point 2: Line 227: “After the first 14 days of cultivation, the expression level of the ScBx1 gene was much higher than that of the ScIgl gene in all inbred lines”à I cannot see this affirmation in the line D39, in which I only can see an increase of two fold in the ScBx1 expression with respect to the ScIgl expression. Please, describe it adequately.
Response 2 corrected according to the reviewer's comment and we have added some details; now the sentence is: ”After the first 14 days of cultivation, the expression level of the ScBx1 gene was much higher than that of the ScIgl gene in each inbred lines, almost 12-, 64-, and 2-fold in L318, D33, and D39 respectively.”
Point 3: Figure 3: The letters (stadistic analysis) of the line D39 ScBx1 are missing. Please, correct it.
Response 3: We have removed the letters from the Figure 4 (before Figure 3) as the Kruskal-Wallis test did not detected any differences between the values of expression levels at subsequent time points in any of inbred lines. The appropriate comment is given in the result description: “Nevertheless, the differences between the expression level of the ScBx1 gene at subsequent time points within each of the three tested lines were statistically not significant (Figure 2)”.
Point 4: Line 262: “In all lines, the profile of BX synthesis usually corresponded with the profile of ScBx1 gene expression up to the 28th dag, with the exception of the second time point (21st dag), when its expression level increased but content of the majority of BXs decreased (Figure 1, 4A–C)”. à I think that this affirmation is not exact since the profile of BX synthesis also corresponded with the profile of ScIgl gene expression from line D39 up to the 28th dag, with the exception of the second time point (21st dag). Please describe it adequately
Response 4: corrected according to the reviewer's comment and we have added the sentence: ”The same relation was observed for ScIgl gene of line D39.”
Reviewer 2 Report
General remarks
Please give a Figure with the pathway showing products and genes involved, information of Benzoxazinoids (BXs) and significance of the study of their production Please provide more context and previous experimentation in the introduction and make clear that this group has several members in common with the group in the study of Groszyk et al. [14] (is this continued study of the same group? ). Please explain clearly how many biological and technical replicates were used in qRTPCR. Please explain in M&M how “relative expression” was calculated and whether expression values were normalized or transformed. Generally, the MW Pfaffl - 2004 method for calculation of R=2^-ΔΔCP is preferred in relative expression studies. In Fig1,2 and 3 Relative expression reads as suppression because R<1. Please make clear how relative expression was calculated. Also the connecting line is hypothetical (and should be omitted) as the actual expression between time points is unknownPlease describe abbreviation when introduced for the first time. Please provide evidence that the assumptions for ANOVA were met and hold true for the current dataset. Consider using a non-parametric test, if assumptions are not valid. Please state clearly which data transformations and normalizations took place. L197 cause and clause are confusing, please reword sentence. L202 results do not support initial assumption that L 318 is the highest BX producer followed by D33 and D39, please comment why. Fig 5 and 6 are missing replication and error bars, what are the confidence levels or SE? Please simplify charts by removing lines, tables and if possible, use less tick marks. Round Y axis numbers to significant decimal, using the same format one each graph. There are parts in the discussion that repeat themselves or come to the same conclusion using different approaches, please short the discussion section and make it more concise. Please also make clear the importance of this work and explain any possible implications.
Author Response
Point 1: Please give a Figure with the pathway showing products and genes involved, information of Benzoxazinoids (BXs) and significance of the study of their production
Response 1:corrected according to the reviewer's comment – figure added
Point 2: Please provide more context and previous experimentation in the introduction and make clear that this group has several members in common with the group in the study of Groszyk et al. [14] (is this continued study of the same group? ).
Response 2: corrected according to the reviewer's comment; we have replaced the sentence:
“Recently, Groszyk et al. [14] showed that the expression of the ScBx1 gene increases under native conditions until the third day after germination and then decreases rapidly, becoming undetectable on the 21st day.”
to: “Recently, a group of Groszyk et al., which included the last 3 authors of this article, showed that the expression of the ScBx1 gene increases under native conditions until the third day after germination and then decreases rapidly, becoming undetectable on the 21st day [14].”
Point 3: Please explain clearly how many biological and technical replicates were used in qRTPCR.
Response 3: corrected according to the reviewer's comment – appropriate information added: “The qRT-PCR reaction was performed in three biological (four plant per replicate) and two technical replications”.
Point 4: Please explain in M&M how “relative expression” was calculated and whether expression values were normalized or transformed. Generally, the MW Pfaffl - 2004 method for calculation of R=2^-ΔΔCP is preferred in relative expression studies.
Response 4: In M&M section we added the information about how “relative expression” was calculated: “Relative quantification of ScBx1 and ScIgl expression level was calculated using ΔΔCt method. The expression level of studied genes was normalized to the expression level of reference gene (HvAct)”.
Point 5: In Fig1,2 and 3 Relative expression reads as suppression because R<1. Please make clear how relative expression was calculated.
Response 5: The relative expression in our experiment means the expression level of studied ScBx1 and ScIgl genes related to the expression level of reference gene which is HvAct (ΔΔCt method). In order to monitor the gene changes over time, but also in each line separately, we didn’t decide to normalized the expression of studied genes to one line (e.g. to L318) or to one day (e.g. 14 dag). So, reads in Fig. 2, 3, and 4 (before Figure 1, 2, and 3) are not suppressed.
Point 6: Also the connecting line is hypothetical (and should be omitted) as the actual expression between time points is unknown
Response 6: corrected according to the reviewer's comment; we have changed the line chart to bar chart. We have also removed the letters from the Fig. 2 (before Fig. 1) as the Kruskal-Wallis test did not detected any differences between the values of expression levels at subsequent time points in any of inbred lines. The appropriate comment is given in the result description: “Nevertheless, the differences between the expression level of the ScBx1 gene at subsequent time points within each of the three tested lines were statistically not significant (Figure 2)”.
Point 7: Please describe abbreviation when introduced for the first time.
Response 7: corrected according to the reviewer's comment
Point 8: Please provide evidence that the assumptions for ANOVA were met and hold true for the current dataset. Consider using a non-parametric test, if assumptions are not valid.
Response 8: As the assumptions of one-way ANOVA are not fully met due to the sample size and the lack of normal distribution (in many cases) we decided to apply a non-parametric, distribution free Kruskal-Wallis test.
Point 9: Please state clearly which data transformations and normalizations took place.
Response 9: corrected according to the reviewer's comment
Point 10: L197 cause and clause are confusing, please reword sentence.
Response 10: Sorry, we don’t understand this comment; in our opinion the sentence is very clear
Point 11: L202 results do not support initial assumption that L 318 is the highest BX producer followed by D33 and D39, please comment why.
Response 11: Line L318 is characterized by the highest BX production, but under field conditions (it was investigated previously). BX content was a criterion for selecting inbred lines for the current analyses (fragment of M&M: “The criterion for selecting inbred lines was the content of BX measured previously after the period of natural vernalization [16]. L318 was characterized as a line with high, D33 with intermediate, and D39 with low BX content (Supplementary Materials: Table SI”). In this study, we have used not vernalized plants grown under controlled conditions; in these conditions line D33 produced BXs at the highest and two remaining lines at lower level.
Point 12: Fig 5 and 6 are missing replication and error bars, what are the confidence levels or SE?
Response 12: In the Figure 6 and 7 (before Figure 5 and 6) expression levels of ScBx1 and ScIgl in rye plants cv. Stach F1, inoculated with virus-induced silenced ScBx1 gene are presented. They are individual plants so we couldn’t have replicates other than technical. According to the reviewer’s comment we add standard deviation in the figure and also the information about replicates in the text.
Point 13: Please simplify charts by removing lines, tables and if possible, use less tick marks.
Response 13: corrected according to the reviewer's comment
Point 14: Round Y axis numbers to significant decimal, using the same format one each graph.
Response 14: corrected according to the reviewer's comment
Point 15: There are parts in the discussion that repeat themselves or come to the same conclusion using different approaches, please short the discussion section and make it more concise.
Response 15: corrected according to the reviewer's comment; we have improved the discussion by removing some repeats. Further shortening would have, in our opinion, negatively impact on its course and clarity.
Point 16: Please also make clear the importance of this work and explain any possible implications.
Response 16: We have slightly changed the last sentence (now it is: “The obtained results showed new aspects of the genetic background of benzoxazinoid biosynthesis and proved that its first stage can be regulated in an alternative way which should lead to a revision of existing views on this subject.”) and we think, that in its current form it clearly indicates the importance of our results. We believe that considering any possible implications would be to speculative.
Round 2
Reviewer 2 Report
The authors have satisfactorily responded to all my comments. The article can be accepted for publication.